# Vesselformer: Towards Complete 3D Vessel Graph Generation from Images

**Chinmay Prabhakar**[*1]                  CHINMAY.PRABHAKAR@UZH.CH

**Suprosanna Shit**[*1,2]                   SUPROSANNA.SHIT@TUM.DE

**Johannes C. Paetzold**[3]                   J.PAETZOLD@IC.AC.UK

**Ivan Ezhov**[2]                      IVAN.EZHOV@TUM.DE

**Rajat Koner**[4]                   KONER@DBS.IFI.LMU.DE

**Florian Sebastian Kofler**[5]       FLORIAN.KOFLER@HELMHOLTZ-MUENCHEN.DE

**Hongwei Bran Li**[1,2]                    HONGWEI.LI@TUM.DE

**Bjoern H. Menze**[1]                  BJOERN.MENZE@UZH.CH

[1] *Department of Quantitative Biomedicine, University of Zurich, Switzerland*

[2] *Department of Computer Science, Technical University of Munich, Germany*

[3] *BioMedIA, Imperial College London, United Kingdom*

[4] *Ludwig Maximilian University of Munich, Germany*

[5] *Helmholtz AI, Helmholtz Zentrum München, Germany*

**Editors:** Accepted for publication at MIDL 2023

## Abstract

The reconstruction of graph representations from images (Image-to-Graph) is a frequent task, especially in the case of vessel graph extraction from biomedical images. Traditionally, this problem is tackled by a two-stage process: segmentation followed by skeletonization. However, the ambiguity in the heuristic-based pruning of the centerline graph from the skeleta makes it hard to achieve a compact yet faithful graph representation. Recently, *Relationformer* proposed an end-to-end solution to extract graphs directly from images. However, it does not consider edge features, particularly radius information, which is crucial in many applications such as flow simulation. Furthermore, *Relationformer* predicts only patch-based graphs. In this work, we address these two shortcomings. We propose a task-specific token, namely radius-token, which explicitly focuses on capturing radius information between two nodes. Second, we propose an efficient algorithm to infer a large 3D graph from patch inference. Finally, we show experimental results on a synthetic vessel dataset and achieve the first 3D complete graph prediction. Code is available at https://github.com/chinmay5/vesselformer

**Keywords:** Transformer, Vessels Graph Generation, Radius Prediction

## 1. Introduction

Extracting graphs of tubular structures is an essential task in medical imaging and pre-clinical results. It involves identifying underlying structures in graphs, such as vessel or neuronal graphs, from various imaging modalities. This image-to-graph extraction is a crucial task for a variety of downstream applications. For example, simulating blood flow in vessels requires complete graph information. In the case of blood vessels, the graph usually consists of nodes, which denote bifurcation points or vessel locations with significant curvature, and edges that correspond to the

---

[*] Contributed equally

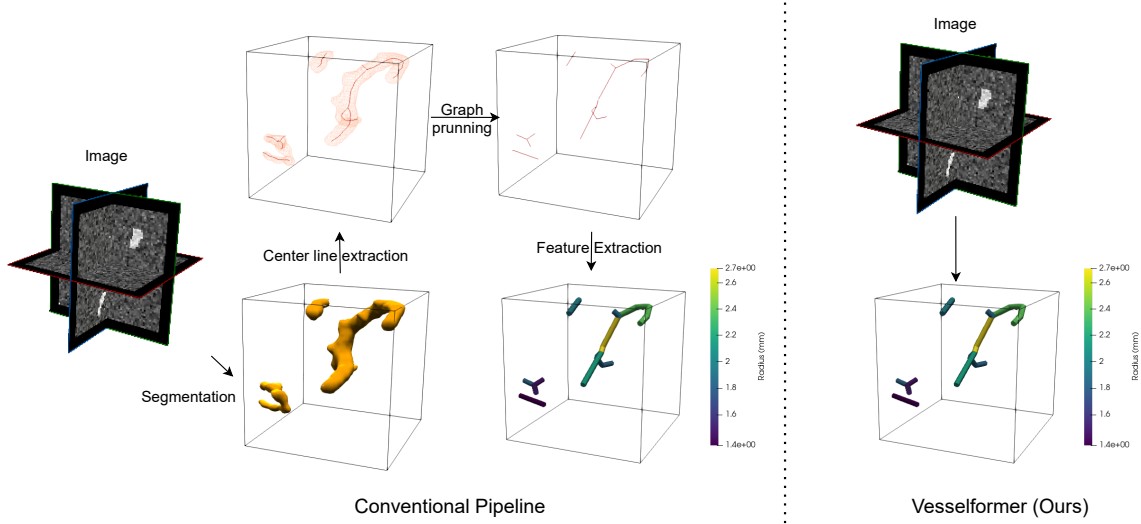

Figure 1: The *Vesselformer* pipeline (right) is shown in comparison with the conventional graph extraction pipeline (left), which involves multiple stages involving segmentation, centerline extraction, graph pruning, and feature extraction. On the other hand, *Vesselformer* is a simple learnable one-stage model to extract compact, representative graphs with features.

vessel segments. Traditionally the image-to-graph problem is broken down into a handful of sub-problems. These problems are typically vessel segmentation, center line extraction, graph pruning, and feature extraction. Not only does this whole pipeline involve many stages it also relies on multiple hand-engineered heuristics. Further, these approaches break the end-to-end differentiability of downstream tasks relying on the inferred graph. Hence a learnable graph extractor is crucial to bridge the gap.

In computer vision, image-to-graph appears in different flavors. The graph structures are semantics instead of structural, as in medical imaging. A recent *Relationformer* method (Shit et al., 2022a) unified these two categories and proposed a general image-to-graph model. *Relationformer* is a transformer-based architecture that operates end-to-end from image to graph. However, *Relationformer* only predicts the structural graph without any features associated with the graph. The edge features, such as radius, in particular, are of paramount importance for vessel applications. The radius information is indispensable for blood flow simulation (Shit et al., 2022b) and vessel graph labeling (Sobisch et al., 2022). Additionally, *Relationformer* generates graphs from small image patches. In different application scenarios, a whole graph of the complete volume, however, is needed in order to facilitate the downstream tasks.

**Our Contribution:** To mitigate these two challenges, we propose *Vesselformer*, a transformer-based model to predict vessel graphs with edge properties such as radius. We achieve this by learning dedicated representation of edge features with the help of proposed [rad]-token in addition to node-to-node similarity measures for edge prediction. Second, we propose an efficient algorithm to combine graphs extracted from image patches into a complete graph for the whole image volume. Finally, we evaluate our method in a synthetic vessel dataset with known radius information.

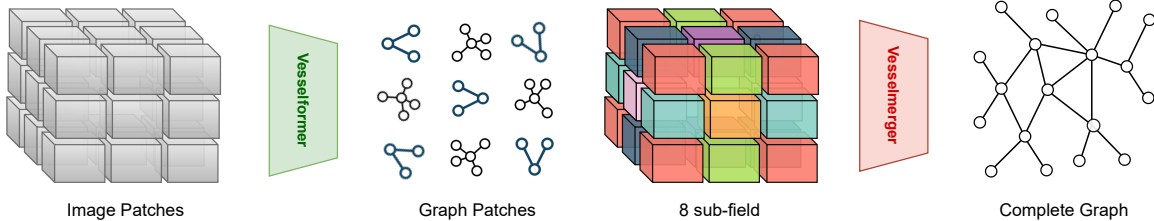

Image Patches          Graph Patches          8 sub-field          Complete Graph

Figure 2: Overview of Vesselformer pipeline, which consists of highly parallelized patch-wise graph-extraction and an efficient merging strategy.

## 2. Related Literature

Despite significant progress in image acquisition and segmentation of curvilinear structures, only a handful of methods exist to extract graphs. Note that all of them are based on hand-engineered rules, and no fully automated solution exists so far. We will discuss a few prominent ones in this section. TubeTK (Aylward and Bullitt, 2002) is an open-access tool, but it only produces vessel segments as tubes rather than a complete graph out of it. Vesselgraph (Paetzold et al., 2021) uses Voreen (Meyer-Spradow et al., 2009), which extracts local properties in each point of centerlines and produces a graph. This, however, reduces to a metric graph, which loses important structural information, such as the twist and curvature of vessels. On a similar note, VesselVio (Bumgarner and Nelson, 2022) proposes a processing pipeline to extract graphs from centerlines. The common problem all skeleton-based methods inherit is dense centerline graphs. There is no good way to prune the graph into a compact yet structurally faithful representation that can be seamlessly transferred to the downstream tasks. In this spirit, our objective is to learn where to place nodes in an image and how to connect two nodes in a data-driven fashion.

Graph extraction from images is a long-standing problem in computer vision that ranges from road network extraction (Xu et al., 2022; Can et al., 2022) or scene-graph extraction (Koner et al., 2020). Most of these modern computer vision image-to-graph extractors are based on a transformer, which leverages the set-prediction formulation to transit from image representation to a discrete graph representation swiftly. Note that all these methods deal with 2D images and are not equipped to tackle increased computational complexity in 3D.

## 3. Methodology

In this section, we will first briefly describe the problem statement. Subsequently, we discuss recent set-based prediction based on transformers, specifically the recent variant, namely *Relationformer*. Next, we will elaborate on our proposed *Vesselformer* framework. Finally, we will explain our proposed graph merging algorithm, namely *Vesselmerger*.

**Problem Statement.** Given an image $I \in \mathbb{R}^{H \times W \times D \times C}$, an image-to-graph task is to predict $\mathcal{G}$, where $\mathcal{G} = (\mathcal{V}, \mathcal{E})$ represents a graph with vertices (or objects) $\mathcal{V}$ and edges (or relations) $\mathcal{E}$. Specifically, the $i^{\text{th}}$ vertex $v^i \in \mathcal{V}$ has a node or object location specified by a location $\boldsymbol{v}^i_{\text{node}} \in \mathbb{R}^3$, a bounding box $\boldsymbol{v}^i_{\text{box}} \in \mathbb{R}^6$ and each edge $e^{ij} \in \mathcal{E}$ has an edge features $e^{ij}_{\text{feat}} \in \mathbb{R}$.

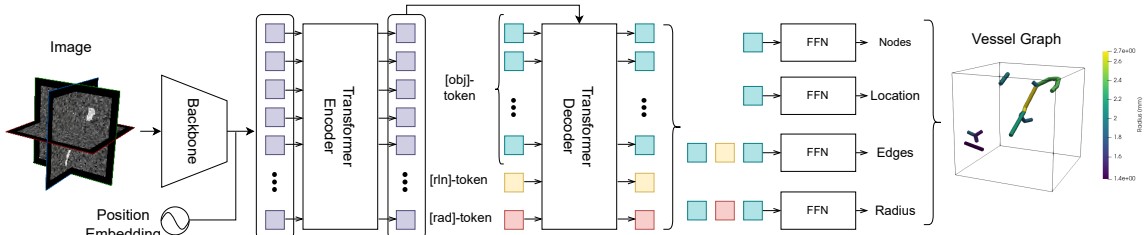

Figure 3: Vesselformer architecture, which learns the node locations in the forms of objects, their similarity in forms of edges, and the edge properties with the help of proposed [rad]-token. Note that the complete pipeline is single-stage end-to-end trainable and simultaneously solves node prediction, edge prediction, and edge attribute prediction.

**Background.** Recently proposed *Relationformer* is a set-based predictor built upon Deformable DETR (Zhu et al., 2021). It relies upon a backbone feature extractor to obtain image tokens. The image tokens are then processed with a series of multi-head self-attention layers. The resultant contextualized image tokens transfer image information to learnable [obj]- tokens and [rln]-tokens through cross-attention. The refined [obj]- tokens produce the node coordinates, and the tuple of two [obj]- tokens and [rln]- tokens produce the final edges through two separate heads consisting of 3 layers multi-layer perceptron (MLP).

### 3.1. Vesselformer

*Vesselformer* originates with the aim of mitigating the shortcomings of *Relationformer*. A key purpose is to integrate edge attributes such as radius prediction. Note that, unlike edge prediction, edge attributes require looking into the image features more thoroughly because radius information is implicit in terms of vessel thickness. Further, edge classification requires solving an affinity measure between two nodes, while radius information is an actual physical quantity. Hence we argue that the edge representation and edge attribute representation lie in different embedding spaces. With this intuition, we propose the following *Vesselformer* architecture.

**Backbone:** We use a 3D fully-convolutional network to extract features from the input image. The final features are flattened to obtain the input sequence for the transformer. A positional embedding is added before feeding the sequence to the transformer.

**Transformer:** We use a transformer encoder-decoder architecture with deformable attention (Zhu et al., 2021), which considerably speeds up the training convergence of DETR by exploiting spatial sparsity of the image features. Our encoder remains unchanged and uses multi-scale deformable self-attention. In the decoder, we introduce a new learnable token in addition to [obj]- tokens and [rln]- token, namely [rad]- token.

**Object Head:** Object detection head consists of two components; one to predict the node location and the other to classify whether the node is valid or background. Additionally, we consider a hypothetical object bounding box of fixed width around the node coordinate. The classification head is trained with cross-entropy loss, while the location head is trained with $\ell_1$ regression and generalized IoU losses.

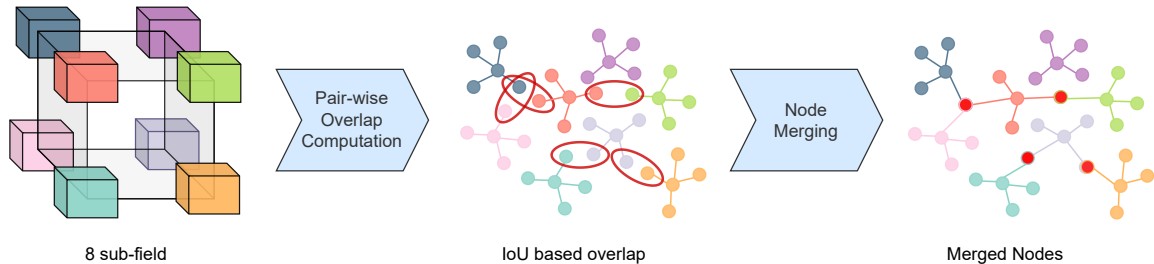

Figure 4: A pictorial demonstration of Vesselmerger algorithm, which heavily parallelizes merging of extracted graphs from each patch into a complete graph output.

**Relation Head:** The relation head is similar to *Relationformer*. This takes the tuple of two object tokens concatenated with the relation token as input and classifies whether an edge exists between two nodes. This head is trained with cross-entropy loss with stochastically sampled edges.

**Radius Head:** We opt for a different embedding space for the radius information and hence employ a dedicated regression head for this task. For the radius head, we use a three-layer MLP, which takes concatenated features of two [obj]- tokens and the [rad]- token and predicts the radius. This head is trained with an $\ell_1$ regression loss.

### 3.2. Vesselmerger

Here we address the important task of combining patched vascular graphs into an entire connected graph. This task is challenging but crucial because the preservation of the entire vascular network is a prerequisite for any whole image or whole system study. In image segmentation, this task is a trivial task because the prediction lies in the same regular grid. In graph space, this does not hold because the node coordinates lie in real numbers instead of integers. Moreover, the patching frequently cuts the vessels, leading to difficult scenarios. Further, for images, the overlapping of patches is easy to handle because of the rectilinear grid. For graphs that have an overlap, it is hard to combine because it requires solving a complex subgraph matching problem. To avoid this extra computation, we stick to non-overlapping patches.

We observe that a non-overlapping patch creates a checkerboard pattern in 3D. Importantly, this can be coloured using eight different subfields shown in eight different colours in Fig. 4. This means adjacent patches will have different colours. This implies that to glue our patch graphs into a big graph, we need to compare adjacent patches instead of matching all nodes to all other nodes. This drastically reduces the computational requirement. Specifically, this means we can efficiently parallelize the pairwise patch merging of different colours. To compute the node proximity for a pair of nodes from two different colours, we leverage efficient voxelwise region growth by invoking grey-scale dilation algorithm. Subsequently, we compute the IoU of the node regions and glue the nodes together, which have an overlap. Finally, we collect all the nodes to be merged and compute the mean node location as a new node. We delete all old nodes to be merged and replace with the new node. Then we adapt the edges accordingly so that the neighbors of the old nodes are now connected to the new node. Note that the edge features remain unchanged in this operation.

---

**Algorithm 1:** Vesselmerger

---

**Input:** $I; I \in \mathbb{R}^{H \times W \times D}$

**Parameters:** $w_p, \tau, m$

**Output:** $\mathcal{G}$

$\{I_i\}_{i=1:n}, \{l_i\}_{i=1:n} \leftarrow \text{Patchlabel}(I, w_p);$

$\{\mathcal{V}_i, \mathcal{E}_i\}_{i=1:n} \leftarrow \text{Vesselformer}(\{I_i\}_{i=1:n});$

/* Initilize variables */

$M_{\text{occ}} \leftarrow \mathbf{0}_{8 \times H \times W \times D};$

$c \leftarrow 0;$

$\mathcal{G}, \mathcal{E}' \leftarrow \phi;$

/* Collect patch-graphs */

**for** $i = 0$ **to** $n$ **do**
    $L \leftarrow \text{Nodelabel}(\mathcal{V}_i, w_p, c);$
    $M_{\text{occ}}[l_i] \leftarrow \text{Greydilation}(L, m);$
    Add $(\mathcal{V}_i, \mathcal{E}_i)$ to $\mathcal{G};$
    $c \leftarrow c + |\mathcal{V}_i|;$
**end**

/* Merge nodes using IoU */

**for** $(i, j) \in {}^8C_2$ **do**
    $M_1, M_2 \leftarrow M_{\text{occ}}[l_i], M_{\text{occ}}[l_j];$
    Add $\text{Nodemerge}(M_1, M_2, \tau)$ to $\mathcal{E}';$
**end**

/* Prune duplicate nodes */

**for** $\mathcal{V}' \in subgraph(\mathcal{E}')$ **do**
    $\mathcal{G} \leftarrow \text{Edgeprune}(\mathcal{V}', \mathcal{G});$
**end**

---

**Procedure** `Patchlabel`$(I, w_p)$

    **for** $i = 0$ **to** $n$ **do**
        $I_i \leftarrow \text{Crop}(I, i, w_p);$
        $l_i \leftarrow i \bmod 8;$
    **end**
    **return** $\{I_i\}_{i=1:n}, \{l_i\}_{i=1:n}$

**Procedure** `Nodelabel`$(\mathcal{V}_i, w_p, c)$

    $L \leftarrow \mathbf{0}_{w_p};$
    **for** $v^j \in \mathcal{V}_i$ **do**
        $L[\boldsymbol{v}^j_{\text{node}}] \leftarrow j + c;$
    **end**
    **return** $L$

**Procedure** `Nodemerge`$(M_1, M_2)$

    $\mathcal{E}' = \phi;$
    **for** $i, j \in U(M_1) \times U(M_2)$ **do**
        **if** $IoU(M_1[i], M_2[j]) > \tau$ **then**
            Add $(i, j)$ to $\mathcal{E}';$
        **end**
    **end**
    **return** $\mathcal{E}'$

**Procedure** `Edgeprune`$(\mathcal{V}', \mathcal{G})$

    $\boldsymbol{v}^{\text{new}}_{\text{node}} = \text{mean}(\boldsymbol{v}_{\text{node}} | v \in \mathcal{V}');$
    Add $v^{\text{new}}$ to $\mathcal{G};$
    Remove $\mathcal{V}'$ from $\mathcal{G};$
    Add $\{(v^{\text{new}}, v) | v \in \text{N}_{\mathcal{G}}(\mathcal{V}')\}$ to $\mathcal{G};$
    **return** $\mathcal{G}$

---

This algorithm is shown and explained in Fig. 4 and the Algorithm 1 (Please refer to Table 4 for description of the notations). The complexity is $\mathcal{O}(nk^2)$ where $k$ is the number of nodes in a patch, and $n$ is the number of patches. Note that the number of nodes in a patch is significantly lower than the total number of nodes in a volume.

## 4. Experiments

We test our method on a publicly available dataset (Tetteh et al., 2020). The dataset consists of synthetic images with corresponding graphs and ground truth radius information. The synthetic data generation concept has been widely used in medical imaging literature (Schneider et al., 2012; Menten et al., 2022; Gerl et al., 2020).

The model is trained for 150 epochs with a batch size of 48. We use AdamW optimizer (Loshchilov and Hutter, 2017) with 1 warmup epoch and an initial learning rate of 0.0001, which was annealed using the polynomial annealing method. A 3D adapted version of squeeze-and-excite network (Hu et al., 2018) is used as the backbone. We use 6 attention heads for the self-attention layer with a hidden dimension of 384. The number of [obj] tokens is fixed at 80 in our experiments while we used a single [rel] token.

Table 1: Quantitative evaluation. We show that Vesselformer succesfully predicts edge attributes such as radius information while predicting the graph. We achieve a high accuracy on the radius without a significant performance drop in edge and node detection score. Further, the Vesselmerger can efficiently produce an accurate graph from the patch graph for the whole 3D volume.

| Resolution | Model | Graph | Topology | | Node Det. | | Edge Det. | | Radius |
|---|---|---|---|---|---|---|---|---|---|
| | | SMD $\downarrow$ | $\beta_0$-error $\downarrow$ | $\beta_1$-error $\downarrow$ | mAP $\uparrow$ | mAR $\uparrow$ | mAP $\uparrow$ | mAR $\uparrow$ | MAE $\downarrow$ |
| Patch-level | U-net+heuristics | 0.01982 | 0.3651 | 0.3577 | 18.94 | 29.81 | 17.88 | 27.63 | N.A. |
| | Relationformer | 0.01107 | 0.0956 | 0.0934 | 78.51 | 84.34 | 78.10 | 82.15 | N.A. |
| | Vesselformer | 0.01260 | 0.1018 | 0.0996 | 76.56 | 82.83 | 76.04 | 80.65 | 0.51 |
| Volume-level | Voreen | 0.03071 | 0.2955 | 0.2766 | 36.17 | 43.35 | * | * | 1.79 |
| | Vesselformer+Vesselmerger | 0.01381 | 0.2188 | 0.2054 | 72.32 | 80.11 | 72.19 | 76.24 | 0.52 |

* denotes missing score because of instability in metric computation.

For evaluation, we report the graph Wasserstein distance (SMD) and mean average precision (mAP) and mean average recall (mAR) for node detection and edge detection. Furthermore, we report Betti number error for topological consistency. We report relative error in $\%$ of Betti-0 (which count the number of connected components) and Betti-1 ( which count circular holes) for the generated graph. For radius prediction, we report the mean-absolute error (MAE).

## 4.1. Main Results

We analyze our quantitative results both at two levels; first, at the patch-level ($64 \times 64 \times 64$), and second, at the whole image level ($300 \times 300 \times 600$). For consistency with prior works, we keep the same hyper-parameters and train/val/test split as reported by the authors in *Relationformer* paper. In Table 1, we report various evaluation metrics on the test dataset. We find that our method extracts the radius information jointly with the graph inference at an almost identical cost. Importantly, this radius information can be helpful in numerous downstream tasks. Importantly, the additional task does not deteriorate the performance of the base tasks of node prediction and edge prediction. We attribute this favorable property to the increased expressive power of our network, especially in terms of the proposed rad-token. Moreover, the performance on the whole 3D volume level is satisfactory and shows globally accurate graph prediction.

Figure 5 shows qualitative results on the test set. The reference graph is depicted in the first row, with their edges colored based on their radius values. The model predictions are plotted in the second row. The model successfully predicts most of the nodes and edges in the scene. Furthermore, the associated colormap depicts that the radius predictions are similar to the ground truth.

## 4.2. Ablation & Sensitivity Analysis

We introduce a special [rad]- token dedicated to predicting the radius information. The inclusion of the extra token leads to slight computational overhead and raises questions regarding its necessity. Here, we study whether a unique token is essential and ablate its optimal number.

**Importance of the [rad]-token:** To answer the first question, we train two models, one with a [rad]- token and another without it. Table 2 shows their comparative analysis. The [rad]-

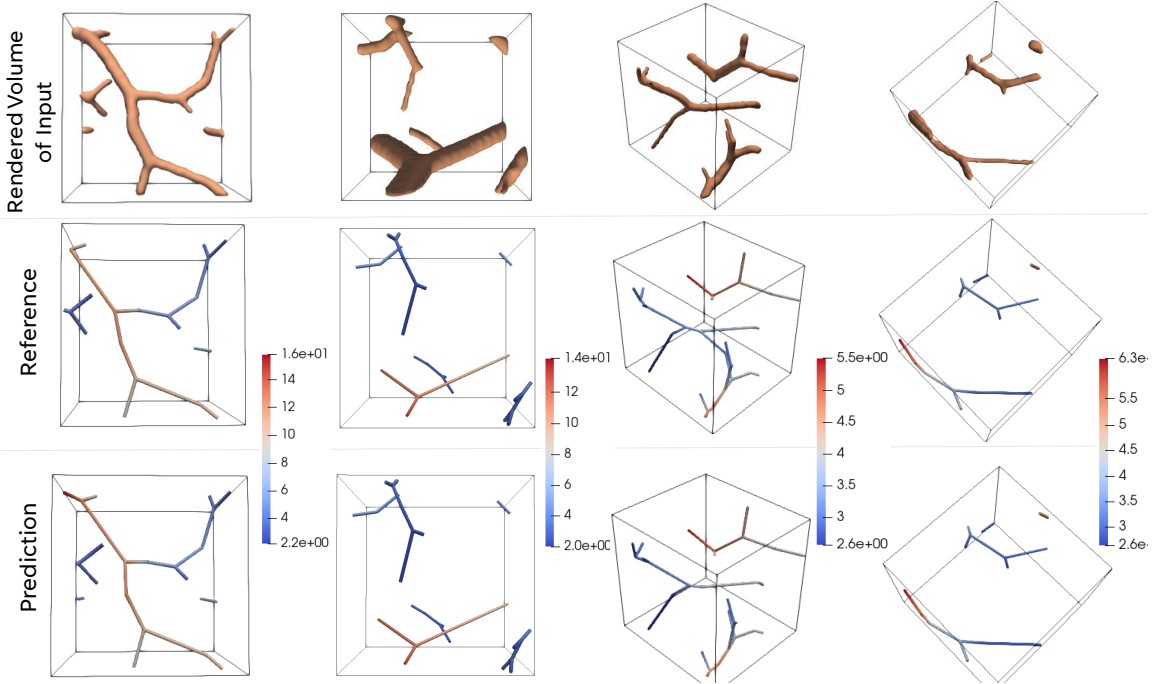

Figure 5: Qualitative results on the test set. The first row shows the ground truth scene while predictions are shown in the second row. Edges are colored based on radius information with the corresponding colormap shown on the right.

token indeed improves the model performance (higher mAP, mAR, and lower SMD, MAE). The mAP and mAR show a sharp drop if we omit the [rad]- token (Table 1). We argue that without a [rad]- token, the onus of learning radius information falls on the [obj]- and [rln]-tokens. This negatively impacts their ability to detect the nodes and edges. The inclusion of the [rad]- token ensures [obj]- and [rln]- tokens can focus on the detection task while radius information can be learned using the dedicated radius head.

**Optimal number of [rad]- token:** Second, we study the optimal [rad]- token number. We trained two models with one and two [rad]- tokens, respectively (Table 1). In both cases, the inclusion of the [rad]- token ensures no catastrophic deterioration in the detection performance. However, the additional [rad]- token leads to a marginal performance drop. Since the additional token leads to computation overhead with no performance gain, we use a *single* [rad]- token in our final model.

**Robustness of Vesselmerger:** We delve into investigating the *Vesselmerger* algorithm in standalone settings. For that, we synthetically generate patch graphs and add Gaussian noise to the normalized node coordinates at the boundary of the patch. Table 3 captures that the *Vesselmerger* algorithm is robust, up to moderate noise level, and hence is a suitable candidate to complement *Vesselformer* for a complete 3D vessel graph.

Table 2: Ablation of `[rad]`-token for Vesselformer. We observe that the absence of the `[rad]`-token reduces performance while increasing its number does not improve performance.

| Model | # `[rad]`-token | Graph SMD ↓ | Topology $\beta_0$-error ↓ | $\beta_1$-error ↓ | Node Det. mAP ↑ | mAR ↑ | Edge Det. mAP ↑ | mAR ↑ | Radius MAE ↓ |
|---|---|---|---|---|---|---|---|---|---|
| Vesselformer | 0 | 0.01260 | 0.1025 | 0.1002 | 76.26 | 82.47 | 75.68 | 80.23 | 0.52 |
| | 1 | 0.01260 | 0.1018 | 0.0996 | 76.56 | 82.83 | 76.04 | 80.65 | 0.51 |
| | 2 | 0.01261 | 0.1021 | 0.0998 | 76.12 | 82.29 | 75.48 | 80.25 | 0.51 |

Table 3: Robustness analysis of Vesselmerger:. We synthetically inject noise of different level in node co-ordinates and observe that Vesselmerger algorithm can tolerate perturbation.

| Model | Noise Level | Graph SMD ↓ | Node Det. mAP ↑ | mAR ↑ | Edge Det. mAP ↑ | mAR ↑ |
|---|---|---|---|---|---|---|
| Vesselmerger | $\sigma = 0.1\%$ | 0.01104 | 94.43 | 98.75 | 90.78 | 93.70 |
| | $\sigma = 0.2\%$ | 0.01275 | 92.23 | 96.57 | 88.34 | 90.86 |
| | $\sigma = 0.5\%$ | 0.01534 | 86.42 | 90.42 | 82.43 | 86.43 |

## 5. Conclusion

In this paper, we addressed the critical topic of vessel graph extraction and provided an efficient learning-based solution. We propose to jointly learn the radius information along with the nodes and edges of the vessel graph. This strategy produces accurate radius prediction with minimal increase in computation. Further, we propose an efficient algorithm to combine graphs from image patches into a whole volume. In summary, we showed that a compact representation of vessels with radius features could be learned with a simple model. We hope future research will strengthen this direction by exploring applications on real large-scale data.

## Acknowledgement

This work was supported by Helmut Horten Foundation and DCoMEX (Grant agreement ID: 956201).

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

## Appendix A. Algorithm Details

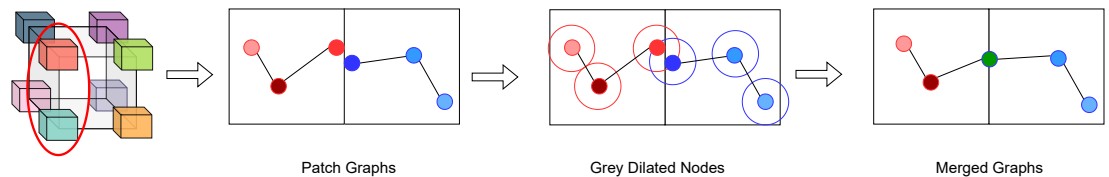

Patch Graphs        Grey Dilated Nodes        Merged Graphs

Figure 6: Pictorial description of Vesselmerger algorithm of two neighboring patches (shown in different colors). Note that the output grey-dilation after multiple iteration helps to identify node overlap across patches. We merge nodes with significant overlap to a new node (shown in green).

Note that at max, we have eight different patches meeting at a coroner, requiring eight different colors to separate them. Nodes lying at each patch's surface (incl. corners and edges) are possible candidates for merging with the similar counterpart of neighboring patches. To identify these candidates, we use the grey dilation algorithm. In grey dilation, we operate on voxel space. The algorithm initializes each node location with a distinct integer value. It places a sphere of unit radius centered on the node location. We execute the algorithm for multiple iterations, and in each iteration, the radius of this sphere is increased. This would lead to a high overlap between adjacent nodes. Next, we compute IoU between the dilated spheres located in two neighboring patches. Finally, we merge the two nodes if the overlap is above a certain threshold. Note that IoU between nodes located in the same patch is not considered, and they do not get merged.

| Notation | Description |
|---|---|
| $M_{\mathrm{occ}}$ | Occupancy matrix for grey dilation |
| $w_p$ | patch size |
| $\tau$ | IoU threshold |
| $m$ | total number of grey dilation iterations |
| $^mC_n$ | m choose n |
| $U(.)$ | Operator returning surface nodes |
| $N_G(.)$ | Operator returning neighboring nodes |

Table 4: Description of the notations used in Vesselmerger algorithm.

## Appendix B. Qualitative Result

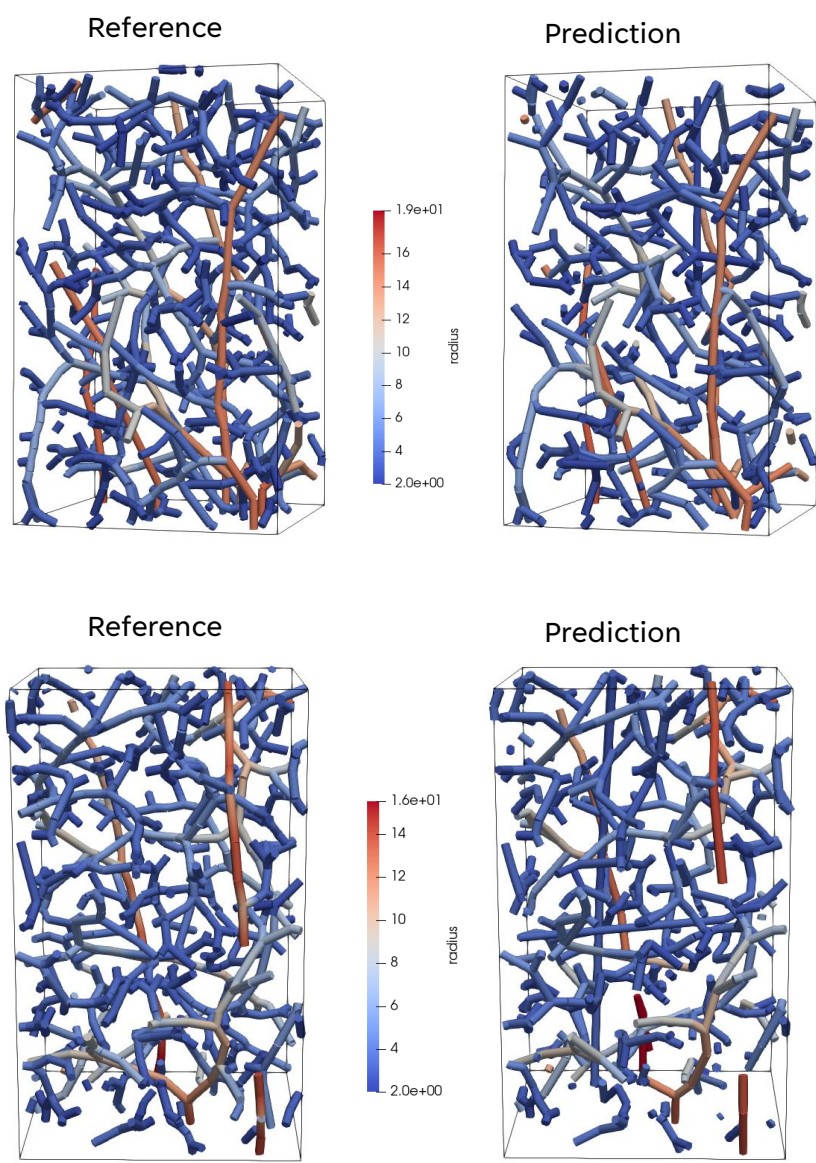

Figure 7: Qualitative results on the volume-level graph extraction on the test set.

Figure 7 shows qualitative results on the test set. We are visualizing the whole graph, obtained after stitching the patches together. The reference graph is depicted on the left. The edges are colored based on their radius values. We have juxtaposed the ground truth and the model predictions. The model successfully predicts most of the nodes and edges in the scene. Furthermore, the associated colormap depicts that the radius predictions are similar to the ground truth.

