# OpenReview forum: "Vesselformer: Towards Complete 3D Vessel Graph Generation from Images"
_MIDL.io/2023/Conference — MIDL 2023 Oral_

### Official Review · Reviewer_Y5q4 · 2023-01-26

**Confidence:** 4
**Preliminary Rating:** 5
**Recommendation:** Oral

**Summary:**

This paper provide a model for 3D vessel graph generation from image, which contains to stages.

In stage one, its Vesselformer uses transformer-based model to build local vessel graph. The transformer encoder-decoder applies deformable attention, which significantly speed up the training. Positional encoding is added to indicate the location of node and edges. Finally, vessel notes, edges and edge radius are predicted combining cross-entropy classification and regression. Multi-task training is applied to output notes, note locations, edges and edge radius.

In stage two, Vesselmerger combines local vessels into a whole one. The merging is based on pre-defined algorithms and non-overlapping local patches.

Both quantitative and qualitative results are provided, indicating the efficiency of this model.

**Strengths:**

The model pipeline involves state-of-the-art deep attention mechanism and transformer encoder-decoders. The efficiency of the Vesselformer relies on the application of these architectures. The algorithm in Vesselmerger algorithm is sophisticated and efficient. Both quantitative result and qualitative exhibition are convincing.

The authors claims that they overcomes the drawbacks of Relationformer, the baseline model from a previous work. Namely, Relationformer cannot predict the radius information of vessel and cannot merge local vessel graphs into a whole one. After reading the paper, these two drawbacks are indeed fixed in the model architecture.

**Weaknesses:**

The author should still provide a quantitative comparison between Relationformer and Vesselformer, maybe not on vessel edge radius, but on other measurements of local graph, such as the location accuracy of vessel nodes. If Vesselformer can indeed out-perform Relationformer on these comparable features or measurements, the paper will be even more convincing.

**Deanonymize Review:**

yes

**Detailed Comments:**

As indicated in the Weakness, the authors should provide quantitative comparisons between Relationformer and Vesselformer.

**Paper Type:**

both

**Questions To Address In The Rebuttal:**

As indicated in the Weakness, the authors should provide quantitative comparisons between Relationformer and Vesselformer. Although this is only a suggestion, I think it is hard for the authors to claim its impossibility.

---

### Official Review · Reviewer_qyDs · 2023-02-03

**Confidence:** 4
**Preliminary Rating:** 5
**Recommendation:** Oral

**Summary:**

This article proposes a learning-based vessel graph extraction approach called Vesselformer.

It is the first learning-based approach that provides a complete 3D vascular graph (enriched with vessel diameters) directly from a 3D vascular image.

This approach is based on the recent Relationformer, which is an end-to-end graph extraction solution that adds relation tokens to the object tokens of deformable detection transformers (DETR) in order to perform both the object detection task and predict the relations between these detected objects.

Whereas Relationformer is a generic solution for directed and undirected graph extraction from 2D and 3D images, Vesselformer is dedicated to the extraction of 3D vascular networks. The authors built upon the Relationformer architecture by adding a radius token to predict the radius of each detected vessel.
The input of Vesselformer is a 3D patch of a 3D volume. In order to produce a final unique vascular graph from all the vascular subgraphs produced by Vesselformer, the authors propose Vesselmerger. Vesselmerger evaluates each set of non-overlapping adjacent patches and merges nodes which are close.

This approach is validated on a synthetic dataset. Experiments are also performed to evaluate the importance of the radius token (presence and number) and the Vesselmerger noise robustness.

**Strengths:**

- The proposed approach tackles a crucial problem encountered in many computer-aided diagnosis tools dealing with vascular diseases. As the authors released their code, it has the potential to be used in numerous clinical applications.
- This is the first learning-based approach that produces a complete 3D vascular graph with associated radius from a 3D image.
- The paper is very well written and illustrated

**Weaknesses:**

- Lack of details to fully understand the Vesselmerger solution (see detailed comments)
- No experiment on real images. The synthetic dataset used in this article is very simple in comparison to real vascular images (high contrast, straight vessels with constant radius…).
- No assessment of the topological correctness of the complete vascular graph

**Deanonymize Review:**

yes

**Detailed Comments:**

This article is very interesting, adapting Relationformer for vascular graph extraction is a good idea and the radius values are indeed crucial information in many applications. Even though the results are promising on the synthetic dataset, the approach should be validated on real 3D vascular images to be convincing. Real vessels are not straight and exhibit a varying diameter and contrast, which makes the problem dramatically more complex.

There is no quantitative evaluation of the topology of the resulting vascular graph. On the qualitative results of the complete graph, the topology of the prediction seems quite different from the reference (many disconnections, circles…). Since preserving the topology of the vascular models is of the utmost importance in many applications, this should be evaluated carefully.

The Vesselmerger approach is not clearly explained. In particular, Algorithm 1 uses many undefined notations (e.g. $M_{occ}$, $w_p$, $m$, $^8C_2$, $U(.)$…). I also do not understand how the dilation is used to select close nodes to merge.

**Paper Type:**

methodological development

**Questions To Address In The Rebuttal:**

- The authors should define all the notations in Algorithm 1
- The authors should explain in more details how they select the nodes that should be merged.  “invoking grey-scale dilation algorithm” is not clear enough.  What is dilated exactly ? Nodes ? reconstructed vessels ? A figure could clarify this point.
- The authors should give information on the computational cost of this approach (e.g. number of epochs, computational time…).
- The authors should quickly explain what they mean by U-net*+heuristics* in the comparison table.
- Please add a visualisation (3D rendering, or 2D slices) of the the initial patch/image in Figure 5.
- Typos
    - Table 1 : Vesselformer+Vesselmeger => vesselmeRger
    - The caption of Figure 5 do not describe the third row
    - End of section 4.1 : Figure 5 shows quantitative ⇒ qualitative

---

### Official Review · Reviewer_Ebqb · 2023-02-03

**Confidence:** 5
**Preliminary Rating:** 2
**Recommendation:** Poster

**Summary:**

This paper suggests a pipeline for fitting tube-graph-structures to 3d images. It is based on a convolution-transformer combination, and it models both the graphs structure and the tube thicknesses. The model is fitted on a publicly available, synthetic data set and it is compared to a single related method from the literature. On these it the proposed model outperforms the method from the literature.

**Strengths:**

The paper is well written, the method is easy to understand, and the results are convincing. I have nothing further to add. I have nothing further to add. I have nothing further to add. I have nothing further to add.


**Weaknesses:**

The paper argues that existing methods rely on "hand-engineered heuristics", however, I find no strong arguments for this work to be any different. I would have liked to se the method tested on real data and compared to more methods. The visualization in Figure 5 is difficult to appreciate due to the sizes and complexities of what is shown.

**Deanonymize Review:**

yes

**Detailed Comments:**

I have no further comments

**Paper Type:**

methodological development

**Questions To Address In The Rebuttal:**

See above. See above. See above. See above. See above. See above. See above. See above. See above. See above. See above. See above. See above. See above. See above. See above. See above. See above. See above.

---

### Meta-Review · Area_Chair_4cbV · 2023-02-21

**Recommendation:** Accept (Poster)
**Confidence:** 5

**Metareview:**

The paper proposes a learning-based approach to extract vessel graphs directly from images. The paper extends RelationFormer to estimate vessel radius information and merge local vessel graphs into a full 3D vessel graph. The paper is well-written and motivated, with a clear clinical utility. However, experiments are only demonstrated on synthetic data, raising concerns about the robustness of the proposed model to complex variations, morphologies, and topologies in real vessels. Nonetheless, the paper is of sufficient interest to the MIDL community and has the potential to advance learning-based vessel extraction methods.